**PLOS** | MEDICINE

**Data Availability Statement:** Daily concentrations for ambient $PM_{2.5}$ and other air pollutants [including sulfur dioxide ($SO_2$), nitrogen dioxide

# Potential gains in life expectancy by attaining daily ambient fine particulate matter pollution standards in mainland China: A modeling study based on nationwide data

Jinlei Qi[1], Zengliang Ruan[2], Zhengmin (Min) Qian[3], Peng Yin[1], Yin Yang[2], Bipin Kumar Acharya[2], Lijun Wang[1]*, Hualiang Lin[2]*

**1** National Center for Chronic and Noncommunicable Disease Control and Prevention, Chinese Center for Disease Control and Prevention, Beijing, China, **2** Department of Epidemiology, School of Public Health, Sun Yat-sen University, Guangzhou, China, **3** College for Public Health & Social Justice, Saint Louis University, St. Louis, Missouri, United States of America

☯ These authors contributed equally to this work.
* wanglijun@ncncd.chinacdc.cn (LW); linhualiang@mail.sysu.edu.cn (HL)

## Abstract

### Background

Ambient fine particulate matter pollution ($PM_{2.5}$) is one leading cause of disease burden, but no study has quantified the association between daily $PM_{2.5}$ exposure and life expectancy. We aimed to assess the potential benefits in life expectancy by attaining the daily $PM_{2.5}$ standards in 72 cities of China during 2013–2016.

### Methods and findings

We applied a two-stage approach for the analysis. At the first stage, we used a generalized additive model (GAM) with a Gaussian link to examine the city-specific short-term association between daily $PM_{2.5}$ and years of life lost (YLL); at the second stage, a random-effects meta-analysis was used to generate the regional and national estimations. We further estimated the potential gains in life expectancy (PGLE) by assuming that ambient $PM_{2.5}$ has met the Chinese National Ambient Air Quality Standard (NAAQS, 75 μg/m³) or the ambient air quality guideline (AQG) of the World Health Organization (WHO) (25 μg/m³). We also calculated the attributable fraction (AF), which denoted the proportion of YLL attributable to a higher-than-standards daily mean $PM_{2.5}$ concentration. During the period from January 18, 2013 to December 31, 2016, we recorded 1,226,849 nonaccidental deaths in the study area. We observed significant associations between daily $PM_{2.5}$ and YLL: each 10 μg/m³ increase in three-day–averaged ($lag_{02}$) $PM_{2.5}$ concentrations corresponded to an increment of 0.43 years of life lost (95% CI: 0.29–0.57). We estimated that 168,065.18 (95% CI: 114,144.91–221,985.45) and 68,684.95 (95% CI: 46,648.79–90,721.11) years of life lost can be avoided by achieving WHO's AQG and Chinese NAAQS in the study area, which corresponded to 0.14 (95% CI: 0.09–0.18) and 0.06 (95% CI: 0.04–0.07) years of gain in life expectancy for each death in these cities. We observed differential regional estimates

($NO_2$), and ozone ($O_3$)] were obtained from the China's National Real-time Publishing Platform for Daily Air Quality (http://106.37.208.233:20035). Daily meteorological data on mean temperature (°C) and relative humidity (%) were obtained from the National Meteorological Data Service Center of China (http://data.cma.cn), which is publicly accessible. The daily mortality time series data were extracted from the Disease Surveillance Points (DSP) System of China, which is operated by the National Center for Chronic and Noncommunicable Disease Control and Prevention, Chinese Center for Disease Control and Prevention, and it cannot be shared according to Personal Information Protection Law in People's Republic of China.

**Funding:** This work was supported by the National Key R&D Program of China (2016YFC0206501) and the Natural Science Foundation of China (81972993); LW and HL received these two awards, respectively. The funders had no role in study design, data collection and analysis, decision to publish, or preparation of the manuscript.

**Competing interests:** The authors have declared that no competing interests exist.

**Abbreviations:** AF, attributable fraction; AQG, air quality guideline; df, degree of freedom; DSP, Disease Surveillance Points; GAM, generalized additive model; GDP, Gross Domestic Product; IT, Interim Target; NAAQS, National Ambient Air Quality Standards; $NO_2$, nitrogen dioxide; $O_3$, ozone; PACF, partial autocorrelation function; PGLE, potential gains in life expectancy; $PM_{2.5}$, particulate matter with an aerodynamic diameter less than or equal to 2.5 μm or fine particulate matter; $SO_2$, sulfur dioxide; STROBE, Strengthening the Reporting of Observational Studies in Epidemiology; WHO, World Health Organization; YLL, years of life lost.

across the 7 regions, with the highest gains in the Northwest region (0.28 years of gain [95% CI: 0.06–0.49]) and the lowest in the North region (0.08 [95% CI: 0.02–0.15]). Furthermore, using WHO's AQG and Chinese NAAQS as the references, we estimated that 1.00% (95% CI: 0.68%–1.32%) and 0.41% (95% CI: 0.28%–0.54%) of YLL could be attributable to the $PM_{2.5}$ exposure at the national level. Findings from this study were mainly limited by the unavailability of data on individual $PM_{2.5}$ exposure.

## Conclusions

This study indicates that significantly longer life expectancy could be achieved by a reduction in the ambient $PM_{2.5}$ concentrations. It also highlights the need to formulate a stricter ambient $PM_{2.5}$ standard at both national and regional levels of China to protect the population's health.

## Author summary

### Why was this study done?

- Ambient fine particulate matter ($PM_{2.5}$) pollution is a severe environmental health concern in China.

- Both short-term and long-term exposure to $PM_{2.5}$ have been found to be associated with increased mortality and years of life lost.

- A few studies have estimated the association between annual $PM_{2.5}$ concentration and life expectancy, but there is no report on the effects of daily $PM_{2.5}$ exposure on life expectancy.

### What did the researchers do and find?

- This nationwide time-series study collected data on more than 1 million nonaccidental deaths in 72 Chinese cities from January 18, 2013 to December 31, 2016.

- We used a generalized additive model to explore the city-specific association between daily $PM_{2.5}$ and years of life lost and then conducted random-effects meta-analyses to generate the regional and national estimates.

- During the study period from January 18, 2013 to December 31, 2016, we estimated that 168,065.18 (about 1.00% of the total) years of life lost can be avoided by achieving WHO's guideline on daily $PM_{2.5}$ concentrations (25 μg/m$^3$) in the study area, which corresponded to 0.14 years of gains in life expectancy for each death.

### What do these findings mean?

- This is the first study to report the potential gains in life expectancy by attaining the daily standards of $PM_{2.5}$, which provided important and useful information of the burden caused by ambient $PM_{2.5}$ pollution.

- In the future, there should be a policy of stricter ambient $PM_{2.5}$ standards at both national and regional levels in China, which would benefit the population's health and life expectancy.

## Introduction

The health effects of fine particulate matter (particulate matter with an aerodynamic diameter less than or equal to 2.5 μm, $PM_{2.5}$) have attracted increasing public concern over the past decade in China [1,2]. The population-weighted annual $PM_{2.5}$ concentration in mainland China reached 54.3 μg/m$^3$ in 2013 [3], which was much higher than that in 1990 (39 μg/m$^3$) and far above the ambient air quality guidelines (AQGs, 25 μg/m$^3$) recommended by the World Health Organization (WHO) [4,5]. Meanwhile, mounting evidence has linked the ambient $PM_{2.5}$ exposure with excess premature deaths and years of life lost (YLL) [6,7]. Such findings have provided valuable information to estimate the disease burden of ambient $PM_{2.5}$ [8,9].

Previous studies have examined the association between short-term and long-term exposure to ambient air pollution and mortality or YLL [10,11]. The short-term studies, usually based on the daily time-series data, evaluated the acute health effects of air pollution [12], while the long-term studies estimated health effects of chronic and cumulative air pollution exposures (usually with the average concentration of several years as the exposure indicator) [13]. The long-term health effect studies generally reported relatively larger effects than those in short-term analyses [14]. The exact biological mechanisms for the health effects of $PM_{2.5}$ exposure remain unclear; previous studies suggested that oxidative stress, systemic inflammation, direct vascular endothelial impairment, and alterations in arterial tone might play important roles [15–17].

Considering the widely reported effects of air pollution exposure on premature mortality and increased years of life lost [7,18], it was reasonable to hypothesize that high levels of air pollution exposure could lead to losses in life expectancy; however, only a few studies have investigated this association, and most of those studies focused on the long-term air pollution exposure [19,20]. For example, one study from the United States and two studies from China reported long-term exposure to higher levels of particulate pollution was associated with reduced life expectancy [21–23]. However, to the best of our knowledge, the evidence is lacking on the effects of short-term (e.g., daily) $PM_{2.5}$ exposure on life expectancy.

Furthermore, there is a need to estimate the potential benefits of reduction in daily ambient $PM_{2.5}$ concentration by attaining the air quality standards. As such, we used potential gains in life expectancy (PGLE) to investigate the benefit on life expectancy by assuming the $PM_{2.5}$ concentration was in compliance with certain ambient air quality standards. Compared with other indicators such as excess mortality and YLL, PGLE is a more informative indicator for epidemiological research [24]. Through directly quantifying the health benefits by attaining the air quality standards, PGLE is more relevant to air pollution controlling and formulation of air quality standards. Another advantage of PGLE is that it can be easily compared across different areas, while excess deaths and YLL are somewhat influenced by the age structure and size of the study population [25]. Although this limitation can be solved by several standardization techniques, the YLL was subject to one important issue of its sensitivity to competing risks of death [25,26].

In this study, we firstly examined the associations between daily $PM_{2.5}$ and YLL after adjusting for potential confounders at both national and regional levels of mainland China from 2013 to 2016, based on which we estimated the PGLE by postulating that ambient $PM_{2.5}$ concentrations were successfully controlled under the Chinese National Ambient Air Quality Standards (NAAQS), as well as WHO's AQG and its Interim Targets (ITs).

## Methods

### Mortality data and YLL calculation

This is a nationwide modeling study based on a time-series analysis. The daily time-series mortality data on nonaccidental causes in 72 Chinese cities (S1 Table) for the period of January 18, 2013 through December 31, 2016 were selected for this study, and a total of 1,226,849 nonaccidental deaths were recorded. The data were extracted from the Disease Surveillance Points (DSP) System of China, which is operated by the National Center for Chronic and Noncommunicable Disease Control and Prevention, Chinese Center for Disease Control and Prevention [27]. The data from the DSP System have been widely used in the assessment of health risk factors or disease burden and policy formulation [28,29]. These cities were selected based on the following process: (1) they were randomly selected using a multistage stratification approach that took the sociodemographic characteristics of the Chinese population into consideration; (2) the daily mortality counts in these cities were temporally stable without large fluctuations, and no change in the administrative divisions occurred during the study period; and (3) their air pollution and meteorological records were accessible during the study period. The completeness and accuracy of the death data in the DSP System were strictly checked by different administrative levels of the Chinese Center for Disease Control and Prevention network. Practitioners in the health facilities were responsible for checking the accuracy, completeness, and data quality of the death data, and they then reported that information to the DSP System. Staff in the district-level CDC reviewed all new information to ensure the data quality (i.e., to check that the ICD codes were maintained and to exclude the duplicate records and redundant information) in the system within 7 days, as well as returning the unclear or uncertain records back to the reporting health facilities. Then, practitioners in those health facilities asked the physicians to correct and confirm the data. Staff in district-level CDC also collected nonaccidental death information from the security department and civil affairs bureau (the other government departments collecting the death information for the purpose of residence) every month. Then, the staff of the provincial- or regional-level CDC would conduct a second round of checking and reviewing. Finally, data were sent to the national-level CDC to undergo a further round of review, which included the duplication, logic, data analysis, and investigation of misreported data.

The 72 cities in our study were divided into the following 7 regions: Northwest, North, Northeast, Central, East, Southwest, and South (Fig 1), and cities in the same region usually incorporated similar features in terms of geographical, meteorological, and cultural conditions.

The average life expectancy in China was 76.25 years in 2016. We used the life expectancy in the corresponding years to calculate the YLL for each death by matching age and sex to the Chinese national life table [30], which was obtained from WHO's website, and then summed the YLL for all deaths on each day of the study period to compute the daily YLL of each city.

This study was based on one project aiming to examine the short-term health effects of air pollution in China, which has been approved by the Ethical Review Committee of Institute for Environmental Health and Related Product Safety, Chinese Center for Disease Control and Prevention. No individual consent was required because all data were analyzed at an

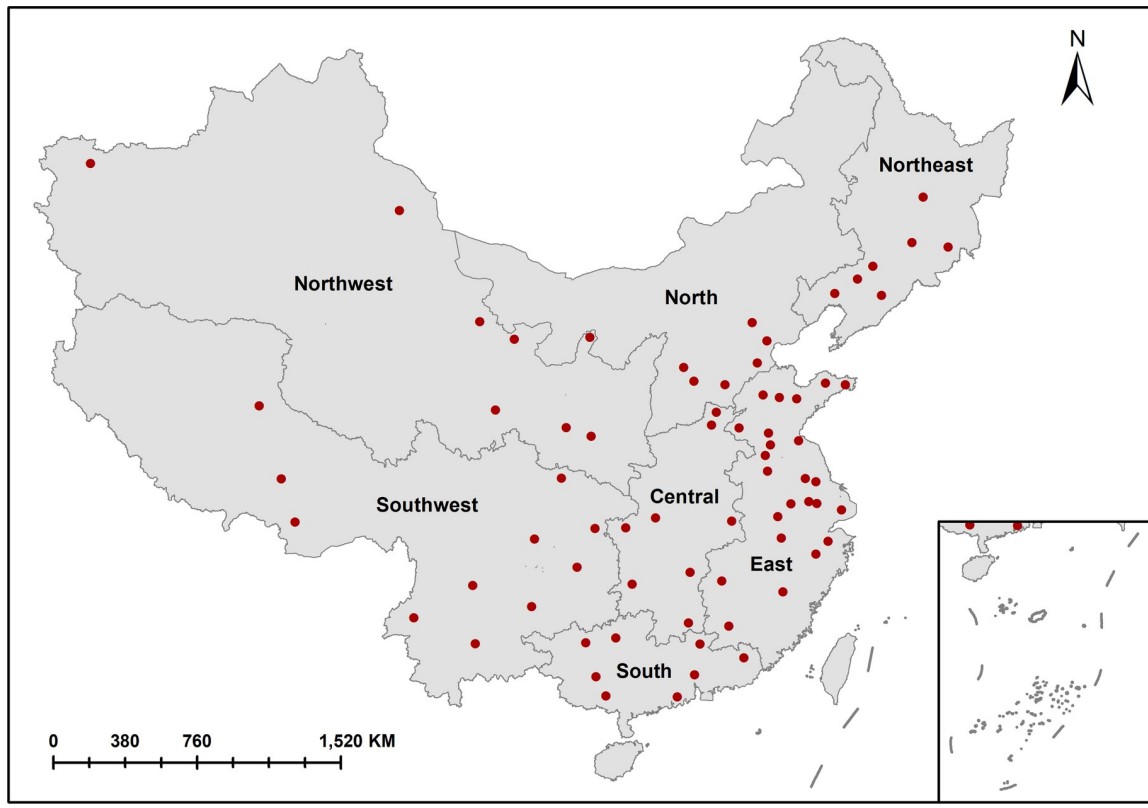

**Fig 1. Geographical distribution of the 72 cities across 7 regions of mainland China.** The location of the 72 cities are indicated by red dots.

aggregated level. The present study is reported as per the Strengthening the Reporting of Observational Studies in Epidemiology (STROBE) guidelines (S1 STROBE Checklist). The data analyses were performed following a prospective analysis plan (S1 Text), and the model structure of this study is provided as a diagram in S1 Fig.

## Air pollution and meteorological factors

Daily concentrations for ambient $PM_{2.5}$ and other air pollutants (including sulfur dioxide [$SO_2$], nitrogen dioxide [$NO_2$], and ozone [$O_3$]) were obtained from China's National Real-time Publishing Platform for Daily Air Quality (http://106.37.208.233:20035), which delivered the real-time concentrations of ambient air pollutants that were measured by state-controlled air-monitoring stations [31]. The 24-hour mean concentrations of ambient $PM_{2.5}$, $SO_2$, and $NO_2$ and the maximum 8-hour mean levels for $O_3$ were averaged from all available monitoring data within each city.

In addition, daily meteorological data on mean temperature (°C) and relative humidity (%) were obtained from the National Meteorological Data Service Center of China (http://data.cma.cn), which is publicly accessible.

## Statistical analysis

**Descriptive analysis.** For descriptive analysis, the number of cities and mean air pollutant concentrations, meteorological conditions, and mortality and YLL in the 7 regions during the study period were summarized. In addition, the Spearman correlation was performed to

quantify the correlation between air pollutants and weather variables. Analyses were based on complete mortality records during the study period.

**Analysis for the PM$_{2.5}$–YLL association.** We examined the national and regional short-term association between daily PM$_{2.5}$ and YLL using two-stage models. At the first stage, we applied a generalized additive model (GAM) with a Gaussian link to explore the city-specific short-term association between daily PM$_{2.5}$ and YLL. In the GAM model, daily mean concentration of PM$_{2.5}$ in each city was incorporated as the independent variable while daily YLL was used as the dependent variable, and all the quantitative variables were treated as continuous variables. We controlled for public holidays and day of the week in the form of categorical variables, while long-term and seasonal trends, temperature, and relative humidity were adjusted using penalized smoothing splines [32]. A complete list of model parameters was provided as a supplemental table (S2 Table). We selected the model specifications and the degree of freedom (df) for the smoothers according to previous experiences of similar studies [33]. For example, we applied a df of 6 per year for long-term trends to filter out the information at time scales of about 2 months, a df of 6 for moving average temperature of the current day and previous 3 days (lag$_{03}$) for the potential nonlinear relationship, and 3 df for the same day's relative humidity. We explored the associations with different lag structures from the current day (lag$_0$) up to 3 days before (lag$_3$), and we also evaluated the effects of moving averages for the current day and the previous 1, 2, and 3 days (lag$_{01}$, lag$_{02}$, lag$_{03}$). The statistical model can be specified as

$$YLL = \beta * PM_{2.5} + s\left(t, df = \frac{6}{year}\right) + \beta_1 * day\ of\ week + \beta_2 * public\ holidays$$
$$+ s(temperature, df = 6) + s(relative\ humidity, df = 3) + \alpha.$$

At the second stage, we used a random-effects meta-analysis to generate the regional and national estimates. This approach provided a useful tool to pool risk estimates while interpreting within-city statistical error and between-city heterogeneity of the genuine risks [34].

**Sensitivity analyses.** We conducted a series of sensitivity analyses to check the robustness of the findings. Two-pollutant models were used to examine the associations between daily PM$_{2.5}$ and YLL after adjusting for other air pollutants. Specifically, PM$_{2.5}$ was included alone in the single-pollutant models, while PM$_{2.5}$ and SO$_2$ (or NO$_2$, O$_3$) were included simultaneously in the two-pollutant models. In addition, we observed that the Northwest and Southwest regions covered a rather large area, which may have wide variation in basic characteristics, and relatively fewer cities were included in these two regions. Considering the uncertainty and the complex geospatial correlation between the cities, we performed a spatial statistical model by adjusting for the longitude and latitude of the cities in the model using a penalized smoothing splines function [35]. Furthermore, we also used a mixed-effect GAM as a one-stage approach to examine the regional and national estimates, in which we included the variable of city as a random term.

In addition, we performed a meta-regression to evaluate whether the observed PM$_{2.5}$–YLL relationship could be explained by some city-level variables: Gross Domestic Product (GDP), population density, GDP per capita, elevation, precipitation, poverty, education, annual PM$_{2.5}$ concentration, annual CO concentration, annual O$_3$ concentration, annual SO$_2$ concentration, annual NO$_2$ concentration, air pressure, annual temperature, and annual relative humidity. The potential interaction between annual PM$_{2.5}$ and GDP was checked by including an interactive term of PM$_{2.5}$ and GDP in the meta-regression model.

**Estimating the avoidable YLL, PGLE, and attributable fraction.** Based on the established associations between ambient PM$_{2.5}$ and YLL, we further estimated the avoidable YLL by assuming the ambient PM$_{2.5}$ had been controlled at specified concentrations as in China's

NAAQS or WHO's AQG and its ITs. We further estimated the PGLE, which was the average years longer each deceased person would have lived if ambient $PM_{2.5}$ were kept under a certain standard in the study area. We also calculated the attributable fraction (AF) that denoted the proportion of YLL due to a higher-than-standards daily $PM_{2.5}$ concentration. The two indicators can be calculated using the following formulas:

$$PGLE = \frac{Avoidable\ YLL}{Overall\ mortality\ count},$$

$$AF = \frac{Avoidable\ YLL}{Overall\ YLL},$$

where avoidable YLL is the sum of estimated YLL that can be prevented in the study area if ambient $PM_{2.5}$ were kept under a certain concentration, overall mortality count is the total mortality number during the study period, AF is the AF, and overall YLL is the sum of the YLL for all deaths that occurred during the study period. The reference levels of $PM_{2.5}$ included WHO's AQG (25 μg/m$^3$) and its ITs, including IT-1 (75 μg/m$^3$, which was the same as the China's NAAQS), IT-2 (50 μg/m$^3$), and IT-3 (37.5 μg/m$^3$).

Our main analyses were performed using R (version 3.5.1; R foundation for Statistical Computing, Vienna, Austria) with the "mgcv" and "metafor" packages. All statistical tests were two-sided, and values of $p < 0.05$ were considered statistically significant.

## Results

### Descriptive results

During the study period, a total of 1,226,849 nonaccidental deaths were recorded in the 72 cities across the 7 regions of China; 44.0% of the study population were females. The average age of death of the subjects included in this study was 71.72 ± 16.74 years. Table 1 summarizes the number of cities, daily mean air pollutant concentrations, meteorological conditions, daily mean mortality, and YLL of these regions. The daily mean concentrations of $PM_{2.5}$, $SO_2$, $NO_2$, and $O_3$ ranged from 49.29 to 95.90, 27.15 to 53.94, 26.80 to 48.99, and 66.75 to 100.69, respectively. The mean temperature ranged from 7.35°C to 21.84°C, and relative humidity ranged from 49.31% to 77.79%. Moreover, the daily mean YLL were 87.16, 123.44, 146.09, 145.04, 158.06, 121.37, and 179.78 years in the 7 regions of Northwest, North, Northeast, Central, East, Southwest, and South, respectively.

The correlation analyses showed low to moderate correlation coefficients between air pollutants and weather variables. For example, $PM_{2.5}$ had moderate positive correlations with $NO_2$ (correlation coefficient = 0.50), had relatively lower correlations with $SO_2$ and $O_3$ (correlation coefficients of 0.26 and 0.29, respectively), and had a negative correlation with mean temperature and relative humidity (correlation coefficients of −0.15 and −0.02, respectively) (S3 Table).

### The association between daily $PM_{2.5}$ and YLL

S2 Fig, S3 Fig, and S4 Fig show the diagnostic graphs of the model, including the plot of the residuals, the plot of partial autocorrelation function (PACF), and Q-Q plot for 6 provincial capital cities. These results showed that there were no discernible autocorrelation and patterns in the residuals, suggesting that the models had acceptable goodness of fit.

In the single-pollutant models, we observed statistically significant associations between $PM_{2.5}$ and YLL at both national and regional levels, especially in the lag$_{02}$ models (Fig 2). At the national level, we estimated that each 10 μg/m$^3$ increase in the $PM_{2.5}$ concentrations of

**Table 1. Summary characteristics of the study cities by regions.**

| Variable | Northwest | North | Northeast | Central | East | Southwest | South | National |
|---|---|---|---|---|---|---|---|---|
| Number of cities | 8 | 6 | 7 | 8 | 24 | 11 | 8 | 72 |
| Mean concentration of air pollutants ($\mu g/m^3$) | | | | | | | | |
| $PM_{2.5}$ | 49.29 | 95.90 | 64.47 | 71.27 | 71.05 | 52.82 | 51.94 | 67.65 |
| $SO_2$ | 34.20 | 53.94 | 45.82 | 31.50 | 41.60 | 28.76 | 27.15 | 38.97 |
| $NO_2$ | 26.80 | 48.99 | 39.96 | 34.99 | 37.24 | 29.73 | 32.02 | 36.70 |
| $O_3$ | 77.72 | 100.69 | 87.33 | 66.75 | 92.76 | 72.28 | 76.04 | 85.14 |
| Weather | | | | | | | | |
| Mean temperature (˚C) | 10.39 | 13.33 | 7.35 | 16.85 | 15.93 | 15.40 | 21.84 | 14.88 |
| Relative humidity (%) | 49.31 | 56.31 | 62.91 | 72.95 | 71.19 | 65.11 | 77.79 | 66.30 |
| Daily mean mortality | 5.32 | 9.68 | 10.20 | 9.77 | 12.39 | 8.06 | 12.89 | 10.34 |
| Daily mean YLL (years) | 87.16 | 123.44 | 146.09 | 145.04 | 158.06 | 121.37 | 179.78 | 141.88 |

**Abbreviations:** $NO_2$, nitrogen dioxide; $O_3$, ozone; $PM_{2.5}$, particulate matter with an aerodynamic diameter less than or equal to 2.5 μm or fine particulate matter; $SO_2$, sulfur dioxide; YLL, years of life lost.

$lag_{02}$ was associated with an increment of 0.43 (95% CI: 0.29–0.57) YLL (S4 Table). The plot of residuals at the national level suggested that these residuals were generally independent, and there were no obviously discernible autocorrelation and patterns (S5 Fig). The region-specific results showed that the associations varied by regions. For example, the Northwest region was found to have the highest association (β = 0.94, 95% CI: 0.21–1.68), while the North region had the lowest association, with a regression coefficient of 0.12 (95% CI: 0.03–0.22).

In the two-pollutant models, the significant associations between $PM_{2.5}$ and YLL generally remained (S4 Table). For instance, at the national level, each 10 μg/m$^3$ increase in $lag_{02}$ $PM_{2.5}$ concentration was associated with an increment of 0.41 (95% CI: 0.27–0.55), 0.32 (95% CI: 0.19–0.45), or 0.41 (95% CI: 0.27–0.55) in YLL after controlling for $SO_2$, $NO_2$, and $O_3$, respectively. The spatial statistical models for the Northwest and Southwest regions, which additionally adjusted for the longitude and latitude of each city, also produced significant effect

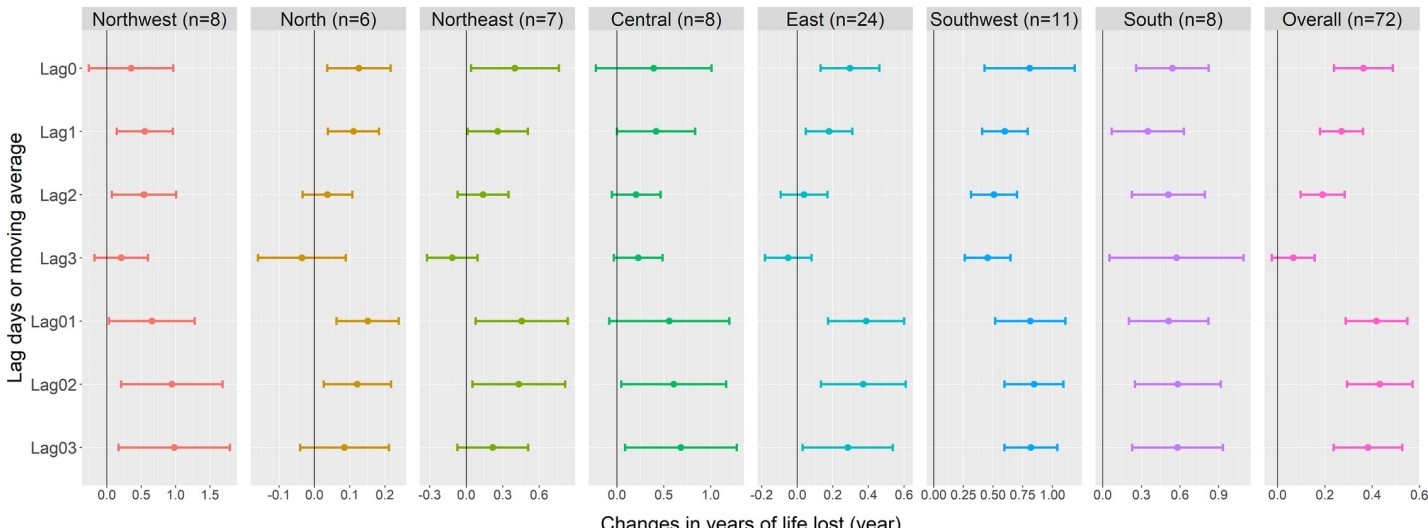

**Fig 2. The associations between ambient $PM_{2.5}$ and YLL in 7 regions of mainland China, 2013–2016.** $PM_{2.5}$, particulate matter with an aerodynamic diameter less than or equal to 2.5 μm or fine particulate matter; YLL, years of life lost.

estimates, with the coefficients of 0.45 (95% CI: 0.31–0.59) for the Northwest region and 0.63 (95% CI: 0.52–0.74) for the Southwest region. While the one-stage mix-effect model also yielded a significant effect (the overall regression coefficient was 0.18 [95% CI: 0.09–0.27]), the estimate was relatively smaller (S5 Table).

In addition, we evaluated whether the observed $PM_{2.5}$–YLL relationship could be explained by some city-level factors (S6 Table). The analysis showed that the associations between $PM_{2.5}$ and YLL were relatively higher in cities with lower annual mean concentrations of $PM_{2.5}$. Each IQR (39.40 μg/m$^3$) increase in annual concentrations of $PM_{2.5}$ was associated with a 0.59 decrease in the regression coefficient. Furthermore, we did not find a significant interactive effect of $PM_{2.5}$ and GDP on the associations between $PM_{2.5}$ and YLL ($p = 0.89$).

### Avoidable YLL, PGLE, and the AF

Based on the established relationship between daily $PM_{2.5}$ and YLL, we estimated the avoidable YLL and AF in different regions of China (Table 2). Specifically, we estimated that 68,684.95 (95% CI: 46,648.79–90,721.11) YLL can potentially be avoided by attaining China's NAAQS (75 μg/m$^3$) in the study area, and this number could rise to 168,065.18 (95% CI: 114,144.91–221,985.45) by meeting WHO's AQG (25 μg/m$^3$). In general, we observed higher effect estimates when adopting stricter air quality standards. Heterogeneity of the estimates was observed across the 7 regions. For example, when adopting WHO's AQG, the East region was found to have the largest avoidable YLL (46,572.93 YLL), while the North region had the lowest avoidable YLL (12,661.48 YLL).

We further estimated that 0.41% (95% CI: 0.28%–0.54%) and 1.00% (95% CI: 0.68%–1.32%) of the YLL could be attributable to the daily exposure of $PM_{2.5}$ by using China's NAAQS and WHO's AQG as the reference (Table 2). In addition, different effect estimates were observed among these regions, with the largest being observed in the Northwest region (1.69% [95% CI: 0.37%–3.02%]) and the minimum in the South region (0.24% [95% CI: 0.10%–0.38%]). Fig 3 shows the regional and national estimates of the PGLE using different air quality standards. Overall, we estimated that 0.14 (95% CI: 0.09–0.18) and 0.06 (95% CI: 0.04–0.07) years in life expectancy can be potentially gained according to WHO's AQG (25 μg/m$^3$) and China's standard (75 μg/m$^3$), respectively. Among the 7 regions, the largest value of 0.28 (95% CI: 0.06–0.49) was observed in the Northwest region, and the minimum value of 0.08% (95% CI: 0.02–0.15) was found in the North region by using WHO's AQG as the reference.

## Discussion

To our knowledge, this might be the first study to quantify the short-term association between ambient $PM_{2.5}$ and life expectancy in China. Using a large data set coving 72 Chinese cities, we estimated that about 0.14 years in life expectancy could be prolonged based on the hypothetical situation that the daily ambient $PM_{2.5}$ concentration was in compliance with WHO's ambient AQG.

Previous studies have well-documented the health effects of ambient air pollutants using a series of health outcomes such as premature mortality, excess morbidity, and YLL, which provided crucial information to measure the harmful effects of ambient air pollutants [36–38]. A few studies further examined the effects of long-term air pollution exposure on life expectancy [39–41]; however, little has been done to address the association of short-term $PM_{2.5}$ exposure with life expectancy, and no studies, to our knowledge, have quantified the potential benefits in life expectancy due to short-term air quality improvement [42,43]. Such evidence will be helpful for policy-making, risk management, and resource allocation.

**Table 2. The avoidable YLL and AF by improving ambient PM$_{2.5}$ to China's and WHO's standards in 72 cities of mainland China, 2013–2016.**

| Region | Avoidable YLL (95% CI) | | AF (%, 95% CI) | |
|---|---|---|---|---|
| | China's Standard (75 μg/m³) | WHO's Guideline (25 μg/m³) | China's Standard (75 μg/m³) | WHO's Guideline (25 μg/m³) |
| Northwest | 4,241.55 (929.63–7,553.46) | 15,911.15 (3,487.29–28,335.01) | 0.45 (0.10–0.80) | 1.69 (0.37–3.02) |
| North | 6,290.23 (1,343.86–11,236.61) | 12,661.48 (2,705.03–22,617.94) | 0.32 (0.07–0.57) | 0.64 (0.14–1.15) |
| Northeast | 7,112.83 (825.47–13,400.18) | 16,660.23 (1,933.49–31,386.97) | 0.39 (0.04–0.73) | 0.91 (0.11–1.71) |
| Central | 7,179.33 (547.35–13,811.32) | 19,483.29 (1,485.40–37,481.17) | 0.43 (0.03–0.82) | 1.16 (0.09–2.23) |
| East | 18,068.94 (6,464.92–29,672.96) | 46,572.93 (16,663.41–76,482.45) | 0.32 (0.11–0.52) | 0.81 (0.29–1.33) |
| Southwest | 12,604.24 (8,889.43–16,319.04) | 33,087.32 (23,335.60–42,839.04) | 0.61 (0.43–0.79) | 1.61 (1.14–2.09) |
| South | 6,261.74 (2,680.24–9,843.24) | 18,909.99 (8,094.13–29,725.86) | 0.24 (0.10–0.38) | 0.72 (0.31–1.13) |
| National | 68,684.95 (46,648.79–90,721.11) | 168,065.18 (114,144.91–221,985.45) | 0.41 (0.28–0.54) | 1.00 (0.68–1.32) |

Based on the effects of moving averaged concentration of lag 0 to lag 2 (lag$_{02}$) of daily PM$_{2.5}$.

**Abbreviations:** AF, attributable fraction; PM$_{2.5}$, particulate matter with an aerodynamic diameter less than or equal to 2.5 μm or fine particulate matter; WHO, World Health Organization; YLL, years of life lost.

A few studies have reported the association between long-term exposure to ambient particulate matter pollution and life expectancy. For example, one study reported that a reduction of 10 μg/m³ in annual PM$_{2.5}$ concentration could increase the life expectancy by about 0.61 years in the United States [21]. Another study similarly reported that an increase of 10 μg/m³ in long-term PM$_{10}$ exposure was associated with a decrease of 0.64 years in life expectancy in China, and it may save 3.7 billion life-years in the whole country if the concentrations of PM$_{10}$ reached the Class I standard of 40 μg/m³ [23]. In the present study, we estimated that 0.14 years in life expectancy can be potentially gained by reaching WHO's AQG on daily PM$_{2.5}$ concentrations in China. This finding was in line with previous observations that the short-term health effects of PM$_{2.5}$ were relatively smaller than those from long-term exposure [44],

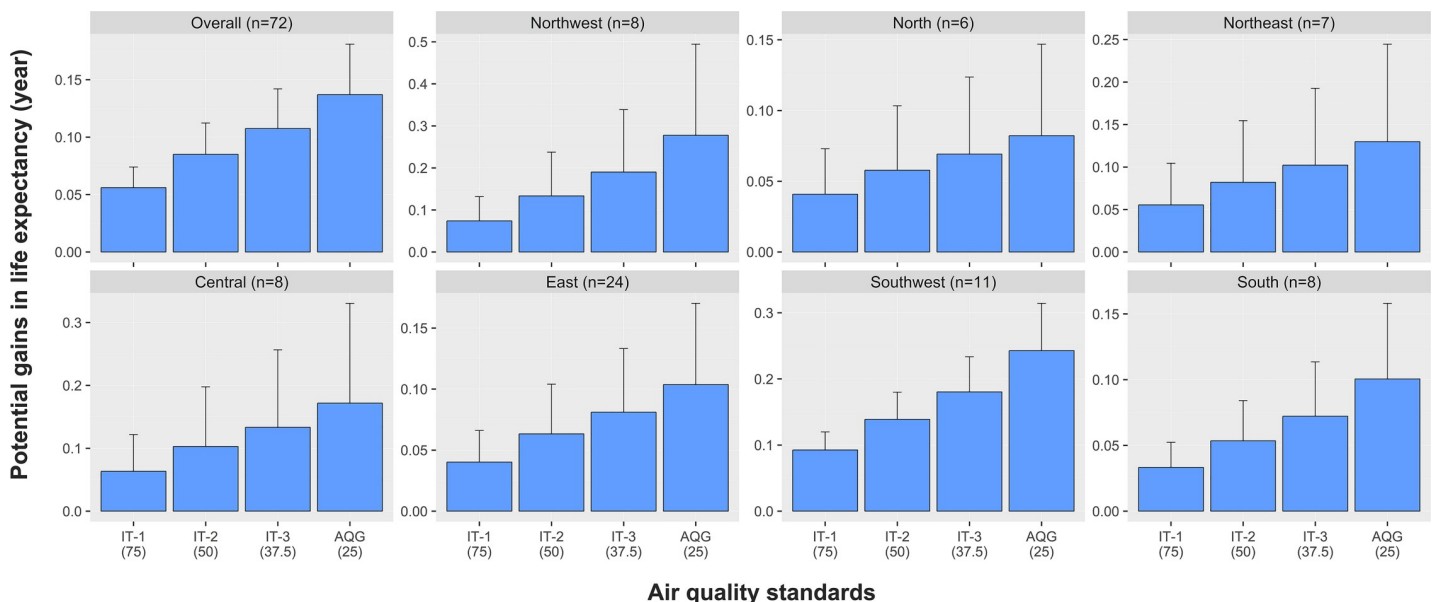

**Fig 3. The estimated PGLE by attaining WHO's AQGs and their ITs in 7 regions of mainland China, 2013–2016.** Based on the effects of moving averaged concentration of lag 0 to lag 2 (lag$_{02}$) of daily PM$_{2.5}$. AQG, air quality guideline; IT, Interim Target; PGLE, potential gains in life expectancy; PM$_{2.5}$, particulate matter with an aerodynamic diameter less than or equal to 2.5 μm or fine particulate matter; WHO, World Health Organization.

and this may be due to the cumulative effects of prolonged exposures [45]. Nevertheless, findings from this study provided valuable evidence for the potential benefits in life expectancy of improved daily air quality, indicating that exposure to higher levels of air pollution even for a short time could reduce life expectancy.

The underlying biological mechanisms linking short-term $PM_{2.5}$ exposure to life expectancy included a range of pathophysiological pathways. For example, one reason was that short-term $PM_{2.5}$ exposure could lead to increased mortality and morbidity of cardiopulmonary diseases through formation of atherosclerotic plaque, systemic oxidative stress, and inflammation [46,47]. This explanation was supported by an intervention study that reduction of particle exposure by indoor air filtration could improve microvascular function in the elderly [48].

We observed a larger potential health benefit when using WHO's AQG (25 μg/m$^3$) as the reference than using China's NAAQS (75 μg/m$^3$), indicating that a stricter ambient quality standard would lead to more health benefits and therefore should be considered in future revision of China's air quality standards.

We observed some evidence for spatial heterogeneity in the association between $PM_{2.5}$ and YLL across different regions. This finding was in line with previous studies [31,49]. Generally, we found relatively weaker associations in the North, East, and South regions, whereas the associations were stronger in the Northwest and Southwest regions. The underlying reasons remained unclear. One possible underlying reason might be the differences in emission sources and chemical constituents of ambient $PM_{2.5}$ among the different regions. The $PM_{2.5}$ in the Northwest and Southwest regions may be more hazardous than that in other regions; most of the ambient fine particles were related to biomass combustion, which was more toxic than other sources [50]. Our meta-regression analysis showed that the areas with higher annual concentrations of $PM_{2.5}$ tended to have a lower $PM_{2.5}$–YLL association, indicating a better adaptation to the local environmental conditions in the areas with higher levels of air pollution. It was possible that people living in highly polluted areas have higher self-protection awareness, which could lead to taking better protective actions such as wearing masks, reducing outdoor activities, and use of air purifiers [31]. Moreover, considering that the cities with a higher $PM_{2.5}$ concentration may also be wealthier and have better healthcare access, we cannot rule out the possibility that there may be some protective effect of economic development level. We therefore included the interactive term of $PM_{2.5}$ and GDP in the meta-regression model and did not find a significant interactive effect. Additionally, in light of previous studies that reported varying effects of $PM_{2.5}$ constituents on human health, we suspect that the differences in chemical components of $PM_{2.5}$ in different areas may be a potential explanation [51,52].

The observed associations between $PM_{2.5}$ and YLL were generally robust in the sensitivity analyses. In particular, the associations remained consistent in the two-pollutant models with adjustment for other air pollutants, indicating that the associations were not confounded by these air pollutants. However, we observed a relatively smaller estimate when adjusting for $NO_2$, which could be partly explained by the moderate positive correlation between $PM_{2.5}$ and $NO_2$ (r = 0.50). It was also possible that $PM_{2.5}$ and $NO_2$ shared similar emission sources and biological pathways in their health effects [53,54]. Furthermore, spatial autocorrelation might be one issue in this analysis; however, this concern should be minimal because the cities were sparsely distributed in different areas, and our spatial model controlling for longitude and latitude of the cities yielded a consistent result.

This study applied a novel, to our knowledge, indicator, namely PGLE, to measure the potential health benefits by controlling air pollution to a certain level. This indicator estimated the average years a person would have lived longer through air quality improvement. This

measurement took into consideration of the age of the deceased and the population size of the study area, making it comparable across different areas [55].

A few limitations should be noted for this study. This was an ecological time-series study, which used the city-averaged concentrations of ambient air pollutants as the exposure measurement. It might have led to ecological fallacy and thus limited our ability of causal inference. However, it is not feasible to measure every participants' exposure directly for such a large-scale study, and this strategy has been widely used in previous time-series studies [56,57]. Relatively fewer cities were included in some regions such as the Northwest and Southwest regions, which might have limited the representativeness of these two regions; however, our sensitivity analyses based on a spatial statistical model produced consistent results, suggesting that the issue did not affect the result estimate to a great extent.

The findings from this study have some important implications for both public health and environment management. We suggest applying this indicator in future efforts. For example, the PGLE can be applied to estimate the effects of other air pollutants on life expectancy, as well as for conducting studies in different populations. The average life expectancy in China was 76.25 years in 2016. The Chinese government released the Healthy China (HC 2030) blueprint in 2016 as a national strategy. One goal of this plan is to increase the average life expectancy to 79 years by 2030. To achieve that goal, a series of action plans were suggested such as health education, diet control, and sufficient physical exercise [58]. In this respect, our study provided some new evidence that the life expectancy can be prolonged by controlling the concentrations of air pollution, and we suggest that this finding should be considered in future policy-making.

In conclusion, this study indicates that ambient $PM_{2.5}$ might be a risk factor for YLL that should not be neglected, and significantly longer life expectancy could be achieved by a reduction in the pollution level.

## Supporting information

**S1 STROBE Checklist. The checklist of STROBE guidelines.** STROBE, Strengthening the Reporting of Observational Studies in Epidemiology.
(DOCX)

**S1 Table. The list of 72 cities in our study.**
(DOCX)

**S2 Table. The list of model parameters in this study.**
(DOCX)

**S3 Table. Spearman correlation between air pollutants and meteorological factors in 72 cities of mainland China, 2013–2016.**
(DOCX)

**S4 Table. Regional-specific estimates of absolute change in YLL associated with each 10 μg/m$^3$ increase in $PM_{2.5}$ in single- and two-pollutant models in 72 cities of mainland China, 2013–2016.** $PM_{2.5}$, particulate matter with an aerodynamic diameter less than or equal to 2.5 μm or fine particulate matter; YLL, years of life lost.
(DOCX)

**S5 Table. Sensitivity analyses for the absolute change in YLL associated with each 10 μg/m$_3$ increase in $PM_{2.5}$ in different models.** $PM_{2.5}$, particulate matter with an aerodynamic diameter less than or equal to 2.5 μm or fine particulate matter; YLL, years of life lost.
(DOCX)

**S6 Table. Change in PM$_{2.5}$–YLL relationship per IQR increase in city-level variables.**
PM$_{2.5}$, particulate matter with an aerodynamic diameter less than or equal to 2.5 μm or fine
particulate matter; YLL, years of life lost.
(DOCX)

**S1 Text. Prospective analysis plan of the current study.**
(DOCX)

**S1 Fig. Diagram of the model structure.**
(TIF)

**S2 Fig. The plot of the residuals for 6 provincial capital cities.**
(TIF)

**S3 Fig. The plot of PACF for 6 provincial capital cities.** PACF, partial autocorrelation function.
(TIF)

**S4 Fig. The normal Q-Q plot for 6 provincial capital cities.**
(TIF)

**S5 Fig. The plot of the residuals for all cities.**
(TIF)

## Author Contributions

**Conceptualization:** Jinlei Qi, Zengliang Ruan, Zhengmin (Min) Qian, Lijun Wang, Hualiang
Lin.

**Data curation:** Jinlei Qi, Lijun Wang.

**Formal analysis:** Jinlei Qi, Zengliang Ruan, Peng Yin, Yin Yang, Hualiang Lin.

**Funding acquisition:** Lijun Wang.

**Methodology:** Zengliang Ruan, Zhengmin (Min) Qian, Yin Yang, Bipin Kumar Acharya,
Hualiang Lin.

**Project administration:** Lijun Wang.

**Resources:** Zhengmin (Min) Qian.

**Supervision:** Lijun Wang.

**Validation:** Jinlei Qi, Peng Yin, Hualiang Lin.

**Visualization:** Jinlei Qi, Zengliang Ruan, Peng Yin, Bipin Kumar Acharya.

**Writing – original draft:** Zengliang Ruan, Hualiang Lin.

**Writing – review & editing:** Jinlei Qi, Zengliang Ruan, Zhengmin (Min) Qian, Peng Yin, Yin
Yang, Bipin Kumar Acharya, Lijun Wang, Hualiang Lin.

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
