## [Decision Letter · Decision Letter 0]

22 Aug 2019

Dear Dr. Lin,

Thank you very much for submitting your manuscript "Potential gains in life expectancy by attaining daily ambient fine particulate matter pollution standards in mainland China: a nationwide analysis" (PMEDICINE-D-19-01976) for consideration at PLOS Medicine. 

Your paper was evaluated by a senior editor and discussed among all the editors here. It was also discussed with an academic editor with relevant expertise, and sent to three independent reviewers, including a statistical reviewer. The reviews are appended at the bottom of this email and any accompanying reviewer attachments can be seen via the link below:

[LINK]

In light of these reviews, I am afraid that we will not be able to accept the manuscript for publication in the journal in its current form, but we would like to consider a revised version that addresses the reviewers' and editors' comments. Obviously we cannot make any decision about publication until we have seen the revised manuscript and your response, and we plan to seek re-review by one or more of the reviewers. 

We expect to receive your revised manuscript by Sep 12 2019 11:59PM. Please email us (plosmedicine@plos.org) if you have any questions or concerns.

We look forward to receiving your revised manuscript. 

Sincerely,

Thomas McBride, PhD

Senior Editor 

PLOS Medicine

plosmedicine.org

1- Did your study have a prospective protocol or analysis plan? Please state this (either way) early in the Methods section.

c) In either case, changes in the analysis—including those made in response to peer review comments—should be identified as such in the Methods section of the paper, with rationale.

2- Thank you for providing information on how readers can access the primary data. In addition to the url for the air pollutant concentrations, please include the url or email contact information for obtaining meteorological data from the National Weather Data Sharing System of China. Additionally, while we understand that you cannot share the daily mortality time series data, please provide contact information (url or email) you used to obtain this data from the Death Surveillance Points System of China.

3- Please place the study design in your title following the colon. We suggest: “Potential gains in life expectancy by attaining daily ambient fine particulate matter pollution standards in mainland China: a modelling study based on nationwide data”

4- Please structure your abstract using the PLOS Medicine headings (Background, Methods and Findings, Conclusions).

5- Please provide the context of why the study is important in the Abstract Background. The final sentence should clearly state the study question.

6- In the Abstract Methods and Findings, please include the total population of the 72 cities, and the exact date range (include day and month) of the study.

7- It would also be helpful to include the average life expectancy and other demographic information (average age, % female) of the population included.

8- In the last sentence of the Abstract Methods and Findings section, please describe the main limitation(s) of the study's methodology.

9- Is it worth specifying in the Abstract Conclusions that these estimates and conclusions are based on short-term air pollution exposure?

10- At this stage, we ask that you include a short, non-technical Author Summary of your research to make findings accessible to a wide audience that includes both scientists and non-scientists. The Author Summary should immediately follow the Abstract in your revised manuscript. This text is subject to editorial change and should be distinct from the scientific abstract. Please see our author guidelines for more information: https://journals.plos.org/plosmedicine/s/revising-your-manuscript#loc-author-summary

11- Please provide the name(s) of the institutional review board(s) that provided ethical approval, and whether informed consent was provided or waived (if waived, by whom).

12- Please include a list of the 72 cities included, perhaps as a supplemental text referenced from the Methods.

13- Please provide a diagram that shows the model structure, including how the disease natural history is represented, the process and determinants of disease acquisition, and how the putative intervention could affect the system.

14- Please provide a complete list of model parameters, including clear and precise descriptions of the meaning of each parameter, together with the values or ranges for each, with justification or the primary source cited, and important caveats about the use of these values noted.

15- Please provide a diagram that shows the model structure.

16- In the Discussion, please include a paragraph just preceding the Conclusion that discusses implications and next steps for research, clinical practice, and/or public policy.

Comments from the reviewers:

Reviewer #1: The study examined the associations between ambient PM2.5 and years of life lost (YLL) using four-year daily death data (1,226,849 non-accidental deaths) in 72 cities in China. These authors thus estimated the potential gains in life expectancy (PGLE) under the Chinese Ambient Air Quality Standards and WHO's air quality guidelines. They found that ambient PM2.5 reduction was significantly associated with decreased YLL and suggested that WHO's AQG should be selected as reference in China. 

This paper addresses important public health issues related to the air pollution. Both the research aim and results in the study are important and interesting. However, I have some considerations in methods section:

1) The research divided the 72 cities into the seven regions (Figure 1) and assessed the association between PM2.5 and death at seven regional level. Authors state that same region has a similar features (meteorological, culture condition etc.). However, the northwest and southwest cover a rather big area which may have big variation for the features and the two big regions only include a few cities data. Considering this issue, I worry about the research results in regional level which may not be good representative for the two regions. Authors may consider using Bayesian spatial geo-statistical model to assess the association as the model can incorpate the spatial dependence and uncertainly? 

2) These authors used two-stage Bayesian hierarchical statistical models (Generalized linear model with a Gaussian link and Bayesian hierarchical random-effects meta-analysis) to conduct data analysis. I would suggest authors providing more details of modeling such as model formula, goodness-of-fit of model which will help readers to understand the methods.

3) Have these authors considered the other factors such as smoking and social economic factors at city level at meta-analysis?

4) Line 138 GAM or GLM? 

Reviewer #2: This study estimate the potential benefits on life expectancy by the daily ambient PM2.5 concentration meets the Chinese and WHO air quality standard respectively in both national and regional levels of China. The result provides a useful information for policy making on quantifying the beneficial effects of air pollution reduction. This manuscript is suitable to be published in PLOS Medicine after the following revision is conducted. 

General comments:

Dataset includes concentrations of PM2.5, SO2, and NO2 and meteorological parameters such as temperature and relative humidity. However, very few discussion has been made using those air pollutants and meteorological parameters. Only a simple correlation results are shown in Table S1, so the meaning to use the above dataset in this paper is not clear. If this paper includes those data more deeply analysis and discussion is needed. 

The 72 cities are divided into 7 regions in this study. However, there is almost no discussion on the comparison and interpretation about the different results of PGLE, Avoidable YLL and AF among those regions. 

The English of this manuscript needs to be revised by native speaker. 

Specific comments:

Line 50: "air pollution" is no necessary shown as Keywords.

Line 64-65: Data sources needs to be cited for annual PM2.5 concentration. 

Line 103: What does the "randomly select and extract" mean?

Line 105: How to check the data set needs to be described more detailed. 

Line 195-196: How to interpret this correlation results? What does the slightly positive or negative correlation mean? 

Table 1: The difference of those variables among the 7 regions needs to be interpreted and discussed in main text. I think population is also important factor for health risk assessment. Why there is no data and discussion on population in different regions? 

Line 211: After control NO2 the increment of YYL is smaller than control SO2 and O3. The reason needs to be interpreted.

Line 229-231: The description of "reducing reference standard" is not clear.

Line 246-250: The reason and indication from this different results of YLL in 7 regions needs to be interpreted. 

Reviewer #3: Thank you for the opportunity to review this paper. Lin and colleagues conducted a nationwide study in China evaluating the association between daily PM2.5, years of life lost and further modelled potential gains in life due to achieving targeted national and WHO standards of PM2.5 targets. They have found across China, although with variation in levels of potential gains in life expectancy across regions, that overall significant gains in average life expectancy could be made by further reducing concentration levels. The 2-stage modelling approach is commonly used in ecological studies with the first stage estimating association using a generalised linear model (link function in this case was Gaussian) between PM 2.5 exposure and outcome of YLL. The second stage utilised a random-effects meta-analysis to tool cities and regions to obtain a national average. The authors also explored 3 day lag structure and moving averages, confounding by temperature and humidity. The study is well-conducted and presented. The only major issue I would have liked to have seen more thought given to in the analysis is incorporation other population level covariates which may affect this association - education, urbanicity, smoking, per capita income, unemployment come to mind. 

1) Abstract methods does not mention the method on using adopting a two-stage model to combine city-specific results to estimate the nation-wide mean potential years of life expectancy gained 

2) Abstract methods - would be helpful to indicate what is the recommended level of ambient PM 2.5 by the Chinese National air quality threshold and the WHO threshold. 

3) Abstract conclusions - it's unclear what the authors mean by "PM2.5 might be a non-ignored risk factor" as national and international policy is currently targeting reduction in PM2.5 to reduce premature deaths. 

4) Intro lines 77-81. It would be helpful for the reader to clearly delineate between short-term exposure and long-term exposure PM2.5 and what the suggested mechanism is for reduction of life expectancy and how they may differ from one another (i.e. some literature suggests long-term exposure has bigger effects)

5) Intro lines 89-92. YLL is still very commonly used and has been extended by the WHO recommended DALY measure combining mortality and morbidity. Influence by age size of the population can be solved by age-direct standardisation approaches. 

see https://academic.oup.com/ije/article/48/4/1367/5281229

To me, the main advantage of PGLE isn't the influence of age structure and size as standardisation techniques allow for comparisons across difference areas, but rather the fact that YLL is extremely sensitive to competing risks (that is the exposure of interest contributes to several causes of death). This has not been mentioned in the text. 

6) Methods Line 103 - What is the rationale here for random selection? I would have thought that it would be more advantageous to consider stratified sampling strategy to have a equal coverage across regions. Was this a pragmatic decision due to the number of cities which needed to be reduced or had a priori rationale. 

7) Methods line 167 on two-stage Bayesian hierarchical model. This is an appropriate approach, where in stage 1 GLM with Gaussian link to estimate city specific association between PM2.5 and YLL, then using second stage to include a random-effect terms to pool cities to create regional and national estimates. 

My question related to whether authors also consider a fully specified one-stage models (random study intercept or fixed study-specific intercept; random exposure effect; and fixed study-specific effects for covariate), investigating interaction - and if using a 1-stage this would have impacted the results instead of a 2-stage model.

8) Figure 2 - northwest region (n=8) missing reference line for 0.

9) Figure 3 needs better labelling y axis label missing and x-axis labels shared common labels between vertical panels are not intuitive. Also some bars are missing what I assume are the 95% CIs

[LINK]

---

## [Decision Letter · Decision Letter 1]

7 Nov 2019

Dear Dr. Lin,

Thank you very much for submitting your revised manuscript "Potential gains in life expectancy by attaining daily ambient fine particulate matter pollution standards in mainland China: a modelling study based on nationwide data" (PMEDICINE-D-19-01976R1) for consideration at PLOS Medicine. 

Your revision was evaluated by a senior editor and discussed among all the editors here. It was also sent to the original reviewers. The reviews are appended at the bottom of this email and any accompanying reviewer attachments can be seen via the link below:

[LINK]

In light of these reviews, I am afraid that we still will not be able to accept the manuscript for publication in the journal in its current form, but we would like to consider a further revised version that addresses the reviewers' and editors' remaining comments. Obviously we cannot make any decision about publication until we have seen the revised manuscript and your response, and we plan to seek re-review by one or more of the reviewers. 

We expect to receive your revised manuscript by Nov 14 2019 11:59PM. Please email us (plosmedicine@plos.org) if you have any questions or concerns.

We look forward to receiving your revised manuscript. 

Sincerely,

Thomas McBride, PhD

Senior Editor 

PLOS Medicine

plosmedicine.org

1- Thank you for including an Author Summary. The last point of the What did the researchers do and find? Section (“By adopting the WHO’s guideline as a reference, we estimated that about 1.00% of the years of life lost could be avoided in the study area.”) could use a bit more context or explanation. How is this point different from the point that precedes it? Over what timeframe is this estimate? Perhaps the point can be removed.

2- The Abstract Methods and Findings should also include the results for the Chinese NAAQS projection, quoted around line 376.

3- Line 480 do you mean “neglected” instead of “non-ignored”?

4- Please make sure all refrerences use the "Vancouver" style for reference formatting, and see our website for other reference guidelines https://journals.plos.org/plosmedicine/s/submission-guidelines#loc-references

Comments from the reviewers:

Reviewer #1: I think the manuscript has been improved. However, I have little consideration for the spatial model. These authors stated "A spatial statistical model by adjusting for the longitude and latitude of the cities in the model using a penalized smoothing splines function." I would suggest mapping these residuals from the model and check if these residuals are independence or appear spatial auto-correlation. Some discussion for why need to control the spatial auto-correlation in spatial model is needed.

Reviewer #2: I am appreciated to have a chance to review this paper again. 

The authors have answered my comments sufficiently and revised the manuscript accordingling.

I have no more comment. 

There are many comments about the models from other reviewers who are experts on this field. I am not sure whether the reply and revision is sufficient or not.

If the answer and revision about the method can be acceptable, this paper is suitable to be publish. 

Reviewer #3: Thank you for addressing issues raised by in the reviews and incorporating additional analyses using the suggested methods. With these additional analyses, it's clear now that the primary results are robust (after numerous sensitivity/stress tests). There were some differences (i.e. use of one-stage model resulted in smaller effect sizes but the result retained significance), but the authors were able to detail and interpret these findings appropriately. 

For me, the key strength now of the analyses is the separate meta-regression analyses exploring population level confounders. The interesting highlight of the confounders was the annual concentration being higher in areas had smaller effects between PM2.5 and life years. The authors suggest this could be self-awareness in areas with higher annual levels of pollution. I agree this could play a role but the final suggestion I think would be worthwhile to check is if there an interaction between annual PM2.5 and GDP with more industrialised areas being also wealthier. If so, this could partially explain as well whether there is some protection afforded by being located in an area with better health care access and care. This is a minor point though, so for that reason I recommend proceeding to acceptance of the very interesting and well-conducted analyses.

[LINK]

---

## [Decision Letter · Decision Letter 2]

11 Dec 2019

Dear Dr. Lin,

Thank you very much for re-submitting your manuscript "Potential gains in life expectancy by attaining daily ambient fine particulate matter pollution standards in mainland China: a modelling study based on nationwide data" (PMEDICINE-D-19-01976R2) for review by PLOS Medicine.

I have discussed the paper with my colleagues and the academic editor and it was also seen again by one of the original reviewers. I am pleased to say that provided the remaining editorial and production issues are dealt with we are planning to accept the paper for publication in the journal.

[LINK]

We look forward to receiving the revised manuscript by Dec 18 2019 11:59PM. 

Sincerely,

Clare Stone, PhD

Managing Editor 

PLOS Medicine

plosmedicine.org

Requests from Editors:

Line 162 – please remove data contact from main text. 

Comments from Reviewers:

Reviewer #1: I think the paper has been further improved. I am happy with the current version.

[LINK]

---

## [Editor Report · Decision Letter 3]

20 Dec 2019

Dear Prof. Lin, 

On behalf of my colleagues and the academic editor, Dr. Jonathan A. Patz, I am delighted to inform you that your manuscript entitled "Potential gains in life expectancy by attaining daily ambient fine particulate matter pollution standards in mainland China: a modelling study based on nationwide data" (PMEDICINE-D-19-01976R3) has been accepted for publication in PLOS Medicine. 

PRODUCTION PROCESS

PRESS

PROFILE INFORMATION

Thank you again for submitting the manuscript to PLOS Medicine. We look forward to publishing it. 

Best wishes, 

Clare Stone, PhD

Managing Editor 

PLOS Medicine

plosmedicine.org